# Explaining Temporal Plans with Incomplete Knowledge and Sensing Information

**Yaniel Carreno** [1,2,3], **Alan Lindsay** [2], **Ronald P. A. Petrick** [1,2]

[1] Edinburgh Centre for Robotics, Edinburgh, UK
[2] Department of Computer Science, Heriot-Watt University, Edinburgh, UK
[3] School of Informatics, The University of Edinburgh, Edinburgh, UK
{y.carreno, alan.lindsay, r.petrick}@hw.ac.uk

## Abstract

The challenge of explaining AI solutions is driven by the need for trust, transparency in the decision process, and interaction between humans and machines, which allows the first to comprehend the reasoning behind an AI algorithm decision. In recent years, Explainable AI Planning (XAIP) has emerged to provide the grounds for querying AI planner behaviour in multiple settings, such as problems requiring temporal and numeric reasoning. This paper introduces an analysis of explainability for temporal planning problems that require reasoning about incomplete knowledge and sensing information. We present an approach called Explainable AI Planning for Temporally-Contingent Problems (XAIP-TCPs) that defines a set of interesting questions from the temporal and contingent planning perspective, covering numeric, temporal, and contingent notions in the presence of incomplete knowledge and sensing information. We present an analysis of the main elements required to deliver compelling explanations for a new set of domains motivated by real-world problems.

## 1 Introduction

As planning technology has matured over the years, we have seen its adoption in a growing number of real-world applications (Maurelli et al. 2016; Hastie et al. 2018; Bernardini et al. 2020). This can be attributed to the general applicability of planning tools (Kerschke et al. 2019) and the relative flexibility of the various languages available for representing different types of problems (e.g., classical (McDermott et al. 1998), temporal (Fox and Long 2003)). Automated planners operate over a problem model (consisting of domain properties, actions, goals, cost functions, etc.) that must capture critical constraints about the underlying problem in order for a generated *plan*—a structured collection of actions that transforms the model's initial state into a goal state—to be effective for execution. Planning models for real-world applications can be quite complex, representing numeric and temporal constraints and uncertainty about the world.

An issue of growing concern for AI-based approaches to real-world applications is the explainability of the solution to *end users*—interested parties interacting with the system— and the process that brought it about (Smith 2012). The field of Explainable Planning (XAIP) (Fox, Long, and Magazzeni

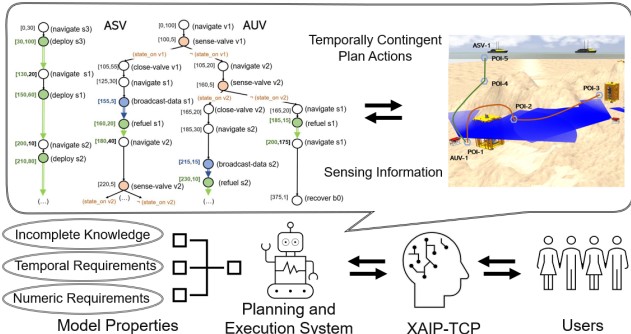

Figure 1: General architecture for plan explainability with temporally-contingent problems. The XAIP-TCP reasoner considers model properties and the output of the planning and execution system to provide explanations to end users.

2017) aims to tackle this problem by considering the need for trust, transparency, and interaction with humans in the context of the planning process. Automated planning solutions are particularly well suited for explanation generation (Chakraborti, Sreedharan, and Kambhampati 2020) due to their use of symbolic models—an approach that has previously been demonstrated by exploiting the planning model to generate explanation content (Chakraborti et al. 2017; Eifler et al. 2020).

Fox, Long, and Magazzeni (2017) presented a general roadmap for XAIP and posed the key questions linking planner behaviour to explainability. The work focuses on plan solutions generated by AI solvers (also known as temporal planners), such as POPF (Coles et al. 2010) and OPTIC (Benton, Coles, and Coles 2012), which solve numeric and temporal constraints. However, the solutions do not consider how they interact with potential uncertainty in the world state. For example, Figure 1 shows a problem where an Autonomous Underwater Vehicle (AUV) has to inspect and close multiple valves with an unknown state (open or ¬open) at planning time. In addition, the AUV needs to refuel during the mission by coordinating with an Autonomous Surface Vehicle (ASV). The ASV is available at different positions in the environment for specific periods. These constraints are considered by the planning and execution system to generate a plan solution that depends on the model properties (e.g., in-

complete knowledge, temporal and numeric requirements). On the planning side, the problem requires the planner to generate multiple sub-plans that deal with all possible valve states. On the user side, an end-user may query the system to ask questions about a generated plan, e.g., Q1: *Why did the planner use action sense-valve?*, or Q2: *What happens if the planner remove action sense-valve?*.

In this paper, we extend the roadmap proposed by Fox, Long, and Magazzeni (2017) to address the main challenges of explainability for Temporally-Contingent Problems (TCPs). We focus on temporal planning problems with numeric constraints where the action sequences required to reach a goal lead to conditional plans resulting from the presence of incomplete information and *sensing actions*. We extend the work in (Smith 2012; Fox, Long, and Magazzeni 2017) by presenting explainability that can help the questioner understand the problem's solution considering the domain's properties represented in the model acquisition (Sreedharan et al. 2020) and plan's output. We (i) introduce the idea of temporally-contingent planning problem, (ii) define a general structure of a plan solution for these planning problems, (iii) introduce two real-world domains that contain temporal and numeric specifications, as well as conditional elements associated with incomplete knowledge and sensing information, and (iv) present Explainable Planning for Temporally-Contingent Problems (XAIP-TCP), defining the main questions and potential answers to explain plans for these type of problems.

## 2 Example Domains

This section introduces two example domains that will motivate our approach and guide the analysis through the paper. These domains[1] are inspired by real-world problems that require both temporal and numeric reasoning to achieve a plan. In addition, domain problems present unknown properties that require sensing actions in the plan to acquire the incomplete/unknown information in the initial states, which leads to planning solutions with multiple branches that respond to all possible outcomes of these unknown properties. Therefore, the use of sensing actions (Petrick and Bacchus 2002; Hoffmann and Brafman 2005; Muise, Belle, and McIlraith 2014) solves the uncertainty in the world state. An example of a branching plan solution in presented in Figure 3, where each plan branch is conditioned on a possible value that a sensing action could return. Plans (including plan branches) may further be required to satisfy certain numeric and temporal constraints. This added complexity in the structure of the plans leads to challenges at the explanation level.

**Domain 1 (Offshore Energy Platform).** This domain is an extension of the Inspection domain in (Carreno et al. 2020b), where a robot has to move to specific locations by choosing among various paths (P-AB1, P-BC1, P-BC2, ...). Figure 2 (left) shows a general representation of the problem, where green arrows indicate possible directions the robot can take from each waypoint. We consider the situation where a robot must reach WP-B and WP-C starting from WP-A. Single or

[1] https://github.com/YanielCarreno/tcp-domains.

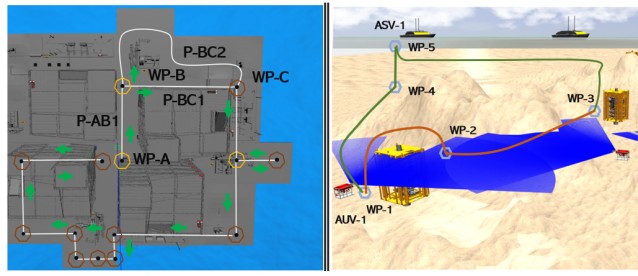

Figure 2: Illustration of the Offshore Energy Platform (left) and the Valve Manipulation (right) domains.

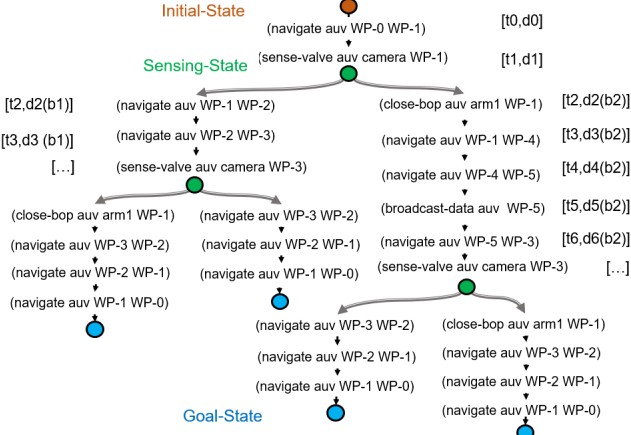

Figure 3: An example of a plan with branches for the Valve Manipulation domain. The outcome of the sensing action `sense-valve` provides certainty around the incomplete knowledge (valve's state) in the initial state. `[t,d]` represents the action start time and its duration respectively.

multiple paths can link points. Robots must use the same paths for navigation, which increases the risk of collision. A robot must observe if the path is clear before travelling a route. The number of routes between points is fixed, with a hierarchy based on distance. We consider that at least one of the paths from a robot's current position is always free.

The domain includes two actions, `navigation(?r ?wpi ?wpf ?p)` and `sense-path(?r ?s ?wpi ?p)`, where `?r` defines the robot, `?wpi` and `?wpf` the initial and final waypoint respectively, and `?s` the sensor. The type `poi` defines the state of the path `?p` between points which is unknown. Therefore, objects of the type `poi` enclose elements of uncertainty that require a sensing action to acquire the incomplete information. A plan solution will present multiple branches after the sensing action that leads to the possible property (path state) outcomes. For this domain, `?p` instances of type `poi` describe the paths. The `navigation` action requires as preconditions to know possible paths between two points and if the path is free. Therefore, `(path_option ?wpi ?wpf ?p)` and `(path_free ?wpi ?p)` have to hold.

WP-A and WP-B are connected by a single path. In this case, the problem initial state defines the single path

connecting the points is known. Therefore, `(path_option wp-a wp-b p-ab1)` and `(path_free wp-a p-ab1)` are at the initial state. Multiple paths exist to navigate from WP-B to WP-C. The first path checked is P-BC1 (shortest path) which is occupied, leading to the sensing action to check other paths. Here, the planner will consider fixed times for sensing action. However, the duration of the navigation can significantly change the plan makespan. Therefore, the implementation of future actions can occur at different time frames depending on the branch. These possible changes in the implementation times have to be considered by the operators to implement further missions that might require coordination amongst a fleet of robots.

**Domain 2 (Valve Manipulation).** This domain represents an updated version of the underwater domains used in (Maurelli et al. 2016; Carreno et al. 2020a). An offshore scenario includes a set of blowout preventers (BOPs), structures with a valve attached that can be open or closed. An AUV must ensure that two valves (v1 and v2) are closed during a mission. The robot has to communicate data every time it is recorded. In the initial state, the robot is at the deployment base. From the base, it is possible to navigate to the BOPs, and from there, the AUV can manipulate the valve. The action the AUV should take depends on the state of the valve: if the valve is open, it should be closed; if the valve is closed, the AUV does not need to perform any action. The valve state can be checked using a sensing action. In addition, the robot needs to refuel during the mission to keep a certain energy level by coordinating a refuel action with an ASV, which is in multiple refuel points at different time slots. Figure 2 (right) shows a general representation of the environment. A plan solution for this problem (assuming no refuelling action) is presented in Figure 3 where, green squares enclosed the sensing actions in the plan. The solution shows a branched plan that consider the possible outcomes of the valve state (unknown property at the planning time) that can be acquired using a sensing action during the plan execution.

In this example the domain action `sense-valve` adds knowledge related to possible valve states. The action `close-bop` includes the precondition `(state_on ?v - poi)`. Therefore, if during plan execution the AUV identifies `(state_on v1)`, the proposition is true along the branch and the action `close-bop` should be planned for the branch the robot takes. Otherwise, if ¬`(state_on v1)` the robot proceeds to execute the next action in the plan associated with a different mission goal. The AI planner needs to reason about possible action sequences considering the sensing action outcomes. Valve Manipulation includes temporal constraints, to support refuelling. In addition, the domain introduces numeric constraints associated with data communication. Action `close-bop` has an effect that increases `(data_acquired ?r - robot)` while action `sense-valve` is conditioned by the robot data capacity. This constraint makes the robot navigate to the surface and communicate data, using action `broadcast-data`, before executing a new sensing action if data was previously acquired. Therefore, the ¬`(state_on ?v)` plan solution presents a completely different sequence of actions.

For this domain, numeric constraints are required to control the data recorded when the valve is turned off. Temporal constraints are essential for scheduling the refuelling activities due to the ASV's time availability at different locations. There is no sequence of actions that allows the AUV to achieve the goal without the knowledge of the valve states: choosing the correct action to execute after sensing the state of a valve depends on the (run-time) result of whether and when the valve is open or ¬open. The characteristics of this problem where the solution requires considering temporal and numeric constraints and reasoning about incomplete sensing information make it a temporally-contingent planning problem.

## 3 Temporally-Contingent Planning

This section defines the TCP model and its plan structure, introducing features defining the problem and guiding the planning search.

**Planning Model.** A common aspect among AI planning approaches is the role of the *model* for finding solutions to a problem. Models define the domain dynamics and are fundamental for obtaining a genuine solution that increases the probability of plan success. Definition 1 describes a temporally-contingent planning problem ($\mathcal{P}_{TC}$). The problem syntax follows PDDL2.1 (Fox and Long 2003) to consider propositional temporal planning problems with Timed Initial Literals (TILs) (Cresswell and Coddington 2003).

**Definition 1.** *A temporally-contingent planning problem is a tuple* $\mathcal{P}_{TC} := \langle \mathcal{P}, \mathcal{F}, \mathcal{A}, \Delta, \mathcal{I}, \mathcal{G}, \mathcal{T} \rangle$*, where* $\mathcal{P}$ *is set of atomic propositions;* $\mathcal{F}$ *is a set of task numeric variables called fluents;* $\mathcal{A}$ *is a set of instantaneous and durative physical actions, with controllable and known duration;* $\Delta$ *is a set of sensing actions (observations), separate from* $\mathcal{A}$ *such that* $\Delta \cap \mathcal{A} = \emptyset$ *that considers temporal notions;* $\mathcal{I}$ *is the set of clauses over the propositions and fluents defining the initial state,* $\mathcal{I} : \mathcal{P} \cup \mathcal{F} \to \{\top, \bot\} \cup \mathbb{R}$*, where* $\top$ *and* $\bot$ *denote the defined and undefined values, respectively;* $\mathcal{G}$ *is a set of goals, where* $\mathcal{G} : \mathcal{P} \cup \mathcal{F} \to \{\top, \bot\} \cup \mathbb{R}$ *is a (possibly partial) function that describes the goal conditions;* $\mathcal{T}$ *is a set of time windows defined as TILs, where each TIL* $l = \langle t(l), lit(l) \rangle \in \mathcal{T}$ *defines the time* $t(l)$ *and the literal* $lit(l)$*, specifying which proposition p becomes true (or false) at time* $t(l)$*, where* $p \in \mathcal{P}$*.*

A literal $l$ is a proposition $p \in \mathcal{P}$ or its negation $\neg p$. A set of literals $\mathcal{L}$ is consistent if the condition $\{p, \neg p\} \nsubseteq \mathcal{L}$ holds; and complete if $\{p, \neg p\} \cap \mathcal{L} \neq \emptyset$ holds for every $p \in \mathcal{P}$. A *state* $s$ is defined as a consistent and complete set of literals. A *belief state* $b$ represents the set of world states that are possible. Actions in the planning model are defined as *physical actions* (see Definition 2) or *sensing actions* (see Definition 3). Sensing actions are nondeterministic actions that can result in more than one possible state.

**Definition 2.** *A (deterministic) instantaneous action* $a_i$*, where* $a_i \in \mathcal{A}$*; and a durative action* $a_d$*, where* $a_d \in \mathcal{A}$*; are defined by the tuple* $\langle a_{i_{pre}}, a_{i_{eff}} \rangle$ *and* $\langle a_{d_{pre}}, a_{d_{eff}}, a_{d_{dur}} \rangle$ *respectively.* $a_{i_{pre}}$ *and* $a_{d_{pre}}$ *represent sets of conditions and preconditions respectively that must hold for the actions to*

*be applicable; $a_{i_{eff}}$ and $a_{d_{eff}}$ represent the sets of action effects; and $a_{d_{dur}}$ is a set of action duration constraints.*

**Definition 3.** *A sensing action $\delta$, where $\delta \in \Delta$ is defined by the tuple $\langle \delta_{pre}, \delta_{eff}, \delta_{dur} \rangle$; $\delta_{pre}$ is the preconditions (a set of literals) required for $\delta$ being executable; $\delta_{eff}$ defines the sensing action effects (a set of literals) where a literal $l$ in the set $\delta_{eff}$ reveals the truth value of the unknown atomic proposition $p \in \mathcal{P}$ at the end of the action; and $\delta_{dur}$ parameter represents a set of duration constraints (controllable and known). Sensing actions are always durative actions.*

The TCP is a special case of Planning in a Partially Observable environment with Sensing actions (PPOS) problems where the source of uncertainty comes from the sensing actions (Muise, Belle, and McIlraith 2014). Our work considers PPOS problems with durative actions. Physical actions are treated as deterministic while sensing actions are encoded as nondeterministic actions. Action `navigation` in the Valve Manipulation domain represents a physical action that has fully deterministic effects, such as the location of the robot `?r`: at the action start (`at ?r ?wpi`), and at the end (`at ?r ?wpf`), therefore, (`explored ?wpf`). Figure 4 shows the sensing action `sense-valve`, which presents an effect associated to define the true state of a valve. The action representation defines the robot `?r` will acquire incomplete knowledge regarding possible values of a particular literal $l$ as an effect of implementing the sensing action. The construct (`at end (K+ (proposition))`) represents this and defines the knowledge acquisition for a proposition with incomplete information. Using this representation, we distinguish the incomplete knowledge associated with a literal $l$ from a false $l$. Here, we present a general structure of the construct, which depends on the designer and introduces the nondeterministic behaviour in the domain. Other authors such as Hoffmann and Brafman (2005) and Petrick and Bacchus (2002) use a different construct with the same objective. Besides, a $\delta$ adds a set of deterministic effects, such as robot `?r` is busy over the action duration, and it is (`available ?r`) when action ends.

Our problems are simple PPOS problems (Bonet and Geffner 2011), which can be mapped into Fully Observable NonDeterministic (FOND) planning problems (Muise, Belle, and McIlraith 2014). For a simple PPOS problem (i) the non-unary clauses in the initial state $\mathcal{I}$ are invariant and (ii) no hidden literal emerges in the effects of a nondeterministic action. Another interesting feature of this type of problem is that uncertainty decreases monotonically, i.e., the unknown properties cannot become unknown again after becoming known. The TCP model encloses the uncertainty associated with the nondeterministic action outcomes. The implementation of *belief tracking*—computation of the successor belief state $b'$—is required to solve the $\mathcal{P}_{TC}$. The TCP model provides the answer to two questions that compile away the uncertainty while searching for a plan. Q1: *What information is unknown/incomplete at the initial state?*, and Q2: *How does the planner reason/update the incomplete knowledge?*.

The first question attempts to define the incomplete knowledge associated with a particular proposition. Follow-

```
(:durative-action sense-valve
 :parameters (?r - auv ?s - sensor ?v - poi ?wp - wpoint)
 :duration ( = ?duration 5)
 :condition (and (over all (at ?r ?wp))
                 (over all (valve_at ?v  ?wp))
                 (over all (camera_equipped ?r ?s)) (...))
  :effect (and (at start (not (available ?r)))
               (at end   (available ?r)) (...)
               (at end   (K+ (state_on ?v)))))
)
```

Figure 4: Durative sensing (PDDL) action `sense-valve`.

```
(:unknown-prop
  (state_on v1)  (state_on v2)
  (...)
  (flow-val v1 f1) (flow-val v1 f2) (flow-val v1 f3)
  (flow-val v2 f1) (flow-val v2 f2) (flow-val v2 f3)
)
```

Figure 5: Construct `unknown-prop` associated to on/off valve's state (top) and possible valve's flows (bottom).

ing the Valve Manipulation domain example, the problem should define the possible (unknown) values (at planning time) the proposition (`state_on ?v`) might hold. Examples of representing incomplete information in the problem model are presented in Figure 5. The first unknown propositions indicate that the (`state_on v1`) and (`state_on v2`) are unknown in the initial state. The second is the case the flow `?f` passing through a valve `?v` is unknown, and the number of (flow) possibilities associated with the same valve is a fixed set. In this case, multiple unknown propositions $p$ represent the possible valve's flow, which differs from the first example associated with a single $p$ (true or false) value. Previous approaches inspire our representation of the incomplete knowledge in the domain (Hoffmann and Brafman 2005; Petrick and Bacchus 2002) that also consider uncertainty in the initial state $\mathcal{I}$.

The second question's answer relates to the generation of the contingent sub-plans (branches) that model the real value of an unknown literal $l$ at the planning time. Figure 6 shows a constructed example that defines the updates associated with the incomplete information that will be true in each branch. The updates are a complex nesting of **and** and **oneof** clauses. Some of the nondeterminism is independent, and others contain dependencies defined by the **and**s. In cases the nondeterminism is independent, the `knowledge-updates` can include information that only branch on *know-whether* facts (e.g., the valve's state is open or closed) that the plan branches on. The incomplete knowledge becomes available at run time through sensing actions, which specify the knowledge's acquisition details. The knowledge acquired is previously defined in the `knowledge-updates` and used at the planning time to generate the set of action's effects associated with the same unknown proposition (e.g., (`state_on ?v`)). The deterministic effects introduced by a sensing action will be true at the end of the action independently of the real value of the

```
(:knowledge-updates
  (oneof (state_on v1)
         (and (not (state_on v1))  (valve_closed wp32)))
  (oneof ... )
  (oneof (and (flow v1 f1)
               (not (flow v1 f2)) (not (flow v1 f3)))
          (and (not (flow v1 f1))
               (flow v1 f2) (not (flow v1 f3)))
          (and (not (flow v1 f1))
               (not (flow v1 f2)) (flow v1 f3)))
)
```

Figure 6: Construct `knowledge-updates` associated to on/off valve's state (top) and possible valve's flows (bottom).

```
Time:   (Action Name)                    [Duration]
  0.00: (navigation auv base v1)           [100.00]
100.01: (sense-valve auv v1)                [30.00]
        <BRANCH, 1, true, (state_on v1)>
130.02:    (close-bop auv v1)               [50.00]
(...)
460.07:    (sense-valve auv v2)             [30.00]
           <BRANCH, 2, true, (state_on v2)>
(...)
           <BRANCH, 2, false, (state_on v2)>
490.08:       (navigation auv v2 base)     [197.67]
687.76:       (recover auv base)             [1.00]
        <BRANCH, 1, false, (state_on v1)>
130.02:    (navigation auv v1 s3)          [80.00]
(...)
```

Figure 7: A temporally-contingent plan solution for a Valve Manipulation domain problem.

unknown proposition. However, deterministic effects could affect the ordering of actions in different branches depending on the true value of an unknown proposition.

A practical example for valve `v1` shows one update will lead to creating a branch that considers (state_on v1) (valve is open). Consequently, action `close-bop` is required in the branch to close the valve and reach the effect (valve_closed wp32), where (valve_at v1 wp32). The second possible update defines the valve as already closed ($\neg$(state_on v1)) and that leads directly to reach the (valve_closed wp32) proposition. In the second example, the `knowledge-updates` establishes a fixed number of possible flows passing through the valve and when one of the flows is true, the others are false (excluded).

**Plan Structure.** The solution to the nondeterministic planning problem with temporal constraints $\mathcal{P}_{TC}$ is thus a contingent plan which induces a set of temporal plans. These problems require combining temporal and contingent planning to deal with observations, incomplete information, temporal and numeric constraints. The solution to this planning problem requires an AI solver capable of solving a time-knowledge aware plan $\Pi_{TC}$ (see Definition 4).

**Definition 4.** *A time-knowledge aware plan $\Pi_{TC} = (\mathcal{N}, \mathcal{E})$ for a temporally-contingent planning problem $\mathcal{P}_{TC}$ is a transition tree $\mathcal{B}$, represented as an AND/OR graph, where nodes $\mathcal{N}$ are labelled with actions built on a set of tuples, $\pi_P := \langle a, t, d \rangle$ for physical actions, and $\pi_S := \langle \delta, t, d \rangle$ for sensing actions; and edges $\mathcal{E}$ represent the action outcomes, denoting the set of propositions whose value are known after an action execution, where $a \in \mathcal{A}$ is an instantaneous or durative action, $\delta \in \Delta$ is a durative sensing action, $t$ is the action starting time, $d$ represents the action duration, $t \in \mathbb{R}_{\geq 0}$, and $d \in \mathbb{R}_{>0}$ when actions have a duration.*

We emphasise $\Pi_{TC}$ solves $\mathcal{P}_{TC}$ iff the executions $\Pi_{TC}$ recommends are applicable in $b$ for $\mathcal{P}_{TC}$ and they lead to a belief state $b''$ where $\mathcal{G}$ holds. The discussion of the AI planner properties is out of the scope of this paper. Figure 7 shows a section of the general structure for a temporally-contingent planning problem solution (plan output) for the Valve Manipulation domain. The contingency elements are represented for the branches, which depends on the effects of the sensing action. The temporal reasoning allows the agent to know the time associated with action implementations for

all contingent sub-plans. Regarding makespan, TCP plan solution introduces the *know-when* concept—property that defines the time $t$ at which the proposition $p \in \mathcal{P}$ knowledge is available—in the plan time-space to specify the time a particular proposition is known. The agent's behaviour is described by one of those plans considering plan execution outcomes. Therefore, the knowledge acquired during plan implementation guides the branch selection.

## 4 Explaining AI Planning for TCPs

This section highlights the main elements of model-based and plan-based explanations and the global connections between these concepts and the TCP. We introduce preliminary features for TCP explainability that are used in Section 5 and Section 6 to explain the specifics of the solutions.

**Model-based Explanation.** Model-based explanations are generated using algorithm-agnostic methods where the model's characteristics support the properties of a solution. Model-based explanation aims to exploit the model to identify properties that can be used to build explanations (Eifler et al. 2020). Explainability approaches based on the model can use two considerations: (i) inference reconciliation; and/or (ii) model reconciliation. For inference reconciliation processes (Zhao and Sukkerd 2019), it is common to allow the users to introduce specific questions about a plan (Fox, Long, and Magazzeni 2017), engage in explanatory dialogue, and/or introduce abstraction techniques that provide the user with tools to understand the plan. In model reconciliation approaches (Chakraborti et al. 2017; Chakraborti, Sreedharan, and Kambhampati 2020) the focus is on the difference between the planner's and the user's models and explanations are generated to align them.

A common point of interest to all approaches involved in explaining AI planning solutions is knowing the model's properties. XAIP-TCP may consider the properties associated with the explanation of deterministic plans, previously examined in (Fox, Long, and Magazzeni 2017; Cashmore et al. 2019) with the nondeterministic elements in the problem. The first reference for the XAIP-TCP to explain a solution involving nondeterministic effects are the model's prop-

erties specified by the Planning and Execution System designer through the questions (see Section 3) that answer the *know-whether* proposition.

**Plan-based Explanation.** A *plan* solution is a fundamental component for XAIP. Planning mechanisms tend to make deterministic and repeatable choices at each decision point. Therefore, the choice of the actions in a plan is transparent at different levels, based on the task's knowledge. The execution of plans generates a sequence of tuples composed of actions, time and observations which can be used: (i) to explore the reasons behind the choice of actions, and (ii) to focus on aspects of state or of action choice, depending on the question to provide explanations. In our work, the plan's output enhances the explanation of observations that lead to multiple sub-plans. These sub-plans are associated with acquiring knowledge or sensing information incomplete at the initial state. Finally, plans support explainability associated with failures. XAIP-TCP can use the *know-when* concept to explain the knowledge acquisition process, which might involve sensing actions. This concept supports plan verbalisation over long-term horizons and large state spaces.

## 5  Questions in the Explanation Process

This section introduces questions to guide the search for explanations of a temporally-contingent planning problem solution. The request for "reasons" explores the available knowledge for the system and is unknown by the questioner. The explanation of a plan should balance the complexity of the (i) reasoning generated by the AI planner and (ii) the question's solution. We focus on inference reconciliation to explain TCP. The formal questions represent an extension to previous work (Fox, Long, and Magazzeni 2017; Eifler et al. 2020) explaining the TCP complexity.

*Q1: Why did the planner use action* ($a$ or $\delta$) *in* $\Pi_{TC}$?. The implementation of an action ($a$ or $\delta$) could be linked to fulfil preconditions required for later actions in the plan that (i) achieve goal states, (ii) maintain resource constraints at optimal levels, or (iii) acquire sensing information.

*Q2: Why did the planner use action* $a$ *in* $\Pi_{TC}$ *after* $\delta$?. The sensing action has the role of acquiring information defined as unknown/incomplete in the initial state $\mathcal{I}$. The selection of $a$ can be linked to the knowledge offered by the execution of $\delta$. However, action $a$ might also be influenced by opportunity and the metric consequences of splitting a plan.

*Q3: Why did the planner not do something different* (at this stage) or (in this branch)?. This question is a version of *Q2*. However, it considers direct alternatives to the initial solution proposed by the planner. This type of question directs the analysis over the alternative plan behaviours the questioner should specify in the question.

*Q4: Why can the planner not do a particular action or sequence?.* This question is associated with the possible unsolvability of planning problems. This type of question tends to be difficult to explain in our domains, considering the plan solution is not completely deterministic. We have a set of conditional sub-plans that we might need to explore (all of

them) to find why the plan fails. These questions can query the domain about specific times for action implementation (particularly sensing actions). For instance, the SATELLITE TIME domain from IPC-4 considers time constraints for the implementation of actions associated to sensing. In our domains, the implementation of a sensing action could depend on numeric constraints.

*Q5: Why is what the planner proposes to do more cost efficient than something else?.* Our domains can present branches with different numbers of actions and sequences that might lead to analyse a wide range of different outputs. This question is very specific to analyse the metric we use for plan evaluation.

*Q6: Why does the planner need to replan if something happens?.* This question focuses on plan execution and intends to analyse the reasons for replanning at particular times. The idea is to look for explainability elements associated with the replanning times and failures. We use the question to find the reasons for replanning related to TCP's characteristics.

## 6  Providing Explanations

In this section, we highlight a roadmap that allows the questioner to clarify the behaviour of an AI planner while solving TCP. We define a set of answers to instances of the questions described in Section 5 based on the domains that motivate this paper. We focus on the tools to approach these questions by analysing the properties of the model or the plan solution. We use two plan solution examples obtained by combining a contingent wrapper and a temporal planner (Carreno, Petillot, and Petrick 2021).

Example 1 (Valve Manipulation): Figure 8 shows the plan solution considering two goals: `(valve_closed wp32)` and `(valve_closed wp34)`.

Example 2 (Offshore Energy Domain): Figure 9 shows the contingent plan solution for two goals: `(inspected pB)` and `(inspected pC)`.

**Explaining Direct Queries.** Causality can explain the need of executing a sensing action $\delta$ early in the plan to support the implementation of action $a$ much later in the same plan. In addition, causality can provide insights into the use of action parameters (e.g., robot, sensor, actuator, etc.). We introduce questions associated with the conditional elements in our model. We use this Example 1 to answer instances of questions *Q1* and *Q2*.

Instance of Q1: *Why did the planner use* `sense-valve` *in the plan solution?* . This question requires an analysis of the causal structure of the plan, including both actions and sensing actions (highlighted in red). The sensing actions provide access to the value of state facts, and the appropriate course of action might be quite different for each sensed value. In this particular example, sensing the valve state is essential for goal completion. As such either `(state_on ?v)` is false and the goal is achieved, or the valve is open and the goal is achieved through the `close-bop` action. However, in some situations, sensing is used to determine the more

```
 Time:    (Action Name)                    [Duration]
 0.00:   (navigation auv base v1)            [100.00]
100.01:  (sense-valve auv camera1 v1)         [30.00]
         <BRANCH, 1, true, (state_on v1)>
130.02:     (close-bop auv v1)                [50.00]
180.03:     (navigation auv v1 surfc.3)       [67.00]
247.04:     (refuel auv surfc.3)              [43.80]
290.85:     (broadcast-data auv surfc.3)      [10.00]
300.86:     (navigation auv surfc.3 v2)       [70.10]
370.97:     (sense-valve auv camera1 v2)      [30.00]
            <BRANCH, 2, true, (state_on v2)>
400.98:        (close-bop auv v2)             [50.00]
450.01:        (navigation auv v2 base)      [160.00]
610.02:        (broadcast-data auv base)      [10.00]
            <BRANCH, 2, false, v2, (state_on)>
400.98:        (navigation auv v2 base)      [260.00]
         <BRANCH, 1, false, (state_on v1)>
130.02:     (navigation auv v1 v2)           [280.00]
410.03:     (sense-valve auv camera1 v2)      [30.00]
            <BRANCH, 2, true, (state_on v2)>
440.04:        (close-bop auv v2)             [50.00]
490.05:        (navigation auv v2 surfc.5)   [140.00]
630.06:        (refuel auv surfc.5)           [42.00]
672.07:        (broadcast-data auv surf.5)    [10.00]
682.08:        (navigation auv surf.5 base)  [320.00]
(...)
```

Figure 8: Temporally-Contingent plan solution for the inspection and manipulation of the valves v1 and v2.

cost-effective route. Existing work on explanations for problems with uncertainty have assumed a policy plan structure, e.g., (Amir and Amir 2018); however, we believe that the structure of the branched plans might be exploited to support plan explanation. A starting point is in the exploitation of visualisations for causal structures (Magnaguagno et al. 2017) and their extension for branching plans.

Instance of Q1: *Why did the planner broadcast at* sufc.3?. For this domain one of the preconditions for manipulation actions is the data_acquired is small than data_capacity. Therefore the basis of explanation in this example might be an analysis of the use of constrained resources in the plan, e.g., (Dvořák and Barták 2010). The answer to this question could be: "The AUV needs to communicate data to free up the data_acquired before executing another δ". However, the reasoning to generate this response is beyond the scope of the paper.

Instance of Q2: *Why did the planner use action* close-bop *for* v1 *after* sense-valve?. The answer to this question is attached to the set of possible states after a sensing action. If the action happens just after a sensing action, the explainability can be based on the branch's information. For instance, an answer to this question can say: "The AUV starts close-bop v1 at 130.02 mins considering v1 state_on". An opportunity indicated by this example is in better approaches for reasoning about and communicating the contribution of a particular (sensing) action to achieving a goal or knowledge gain, perhaps using structures similar to plan property dependencies (Eifler et al. 2020).

Instance of Q2: *Why did the planner use* refuel *at* sufc.5 *after* sense-valve?. Another essential property of TCP plan solutions is they can consider the effect of temporal constraints using TILs. In the Valve Manipulation domain the AUV needs to recharge during the mission. The query relates to the refuel action at time 630.06 mins. The answer to the question follow the same philosophy presented in (Fox, Long, and Magazzeni 2017), and it can be: "The refuel ensures the AUV battery is above the threshold for the subsequent actions in the plan". If we examine the sequence of actions in BRANCH 2, this is the case when valve v1 was closed. Therefore, the AUV does not need to consume battery in closing the valve or communicating data. This allows AUV to move to valve v2 with enough battery to execute the inspection. The question provides information about the branch, therefore we can establish the reasoning comparing the times the refuel action (highlighted in blue) is implemented for each branch which is proportional to the battery consumed.

**Explaining Contrastive Queries.** The contrastive property (Miller 2018) is also considered for XAIP-TCP. (Fox, Long, and Magazzeni 2017) describes the solution to these queries can take into account the number of actions in the "optional" plan (after the introduction of human variations) or the deviation in the goals achieved as a consequence of the changes introduced. However, here we present additional contrastive questions associated to the analysis of an entire branch. Example 1 is linked to *Q3*.

Instance of Q3: *Why did the planner not refuel at* sufc.4 *after closing* v2?. The AUV identifies the valve v1 is open and needs to close it. The robot acquires data from the panel that it needs to communicate. The AUV has to navigate to the surface to execute the communication action. For this solution, the solver reasons the AUV has to reach the surface (to communicate data) and finds a solution that matches with the time the SV is at sufc.3. The explanability for this question can be based on existing approaches in XAIP, such XAIP as a service (Cashmore et al. 2019). For example, a foil can be generated with the added constraint of refuelling at sufc.4. The explanation is then based on the resulting increase in the cost of the resulting plan branches. An interesting possibility presented by the branched plan is to use the different plan branches to provide comparisons, e.g., comparing the locations of the AUV refuels in different branches.

**Explaining Unsolvability Queries.** Unsolvability queries for XAIP-TCPs analyse the failed attempt to implement actions in a given state. The solution to these questions can consider model reconciliation properties (Sreedharan et al. 2019), in cases the action we want to implement prevents the implementation of a goal. The action's properties analysis introduced by the validator VAL (Howey, Long, and Fox 2004) while evaluating the model, in cases, the current state does not satisfy the action precondition. The following two questions connect to *Q4* and Example 1 and Example 2, respectively.

Instance of Q4: *Why can the planner not sense in advance?*.

```
Time: (Action Name)                          [Duration]
 0.00: (navigation husky1 pA pB Path1)           [5.00]
 5.01: (inspect-area husky1 pB)                 [10.00]
15.02: (position-camera husky1 camera1 Path1)    [2.00]
17.03: (sense-path husky1 pB Path1)              [3.00]
       <BRANCH, 1, true, (path_free pB Path1)>
20.04:    (navigation husky1 pB pC Path1)        [8.00]
28.05:    (inspect-area husky1 pC)              [10.00]
       <BRANCH, 1, false, (path_free pB Path1)>
20.04:    (position-camera husky1 camera1 Path2) [2.00]
22.05:    (sense-path husky1 pB Path2)           [3.00]
          <BRANCH, 2, true, (path_free pB Path2)>
25.06:       (navigation husky1 pB pE Path2)    [10.00]
35.07:       (navigation husky1 pE pF Path2)    [12.00]
47.08:       (navigation husky1 pF pG Path2)     [9.00]
58.09:       (navigation husky1 pG pC Path2)     [8.00]
67.01:       (inspect-area husky1 pC)           [10.00]
          <BRANCH, 2, false, (path_free pB Path2)>
```

Figure 9: Temporally-Contingent plan solution for point inspection in an offshore energy simulator environment.

This question opens another set of interesting points around explainability. It suggests the user's model does not capture all of the constraints, and the solution could take inspiration from model reconciliation approaches (Chakraborti et al. 2017). For this particular problem, the sensing actions have to be implemented at a specific position to identify a single state (from a set of possibilities). Therefore, we can approach the question saying: "The AUV needs to be positioned close to the valve to identify its state". In general, answering this question may require extending existing approaches to domains with temporal and numeric constraints, which can support the application of sensing actions.

Instance of Q4: *Why can the planner not navigate Husky1 to* pC *without sensing again at* pB*?*. The plan solution to this domain contains new reasoning associated with the subplans, which must be explained. The sensing action is repeated every time the output from the action sense-path is false. Separate actions are used to represent distinct operations, e.g., the camera needs to face a different direction. Figure 9 shows the case where the maximum number of paths that can be inputs or outputs to a point is two. The explainability could follow the approach to queries: "Why is this not a solution?" in (Sreedharan et al. 2020). The alternative case (where the sensing action at 22.05 mins is prevented) leads to unsolvability.

**Explaining Metric Queries.** For temporally-contingent problems, we consider plan metrics to analyse the quality of the plan solution. The majority of temporal planners base their performance analysis on the plan makespan. This is difficult in contingent planning, where planners ideally optimise the tree as a whole, leading to the higher cost incurred on a particular branch to improve the quality of the overall plan. Understanding the dependencies between the costs in different branches of a plan could provide useful insights for an explanation, e.g., examining the impact when sub-plans have additional cost limits. As a starting point, we consider comparisons amongst alternative branches. The following

question represents an instance of *Q5* for Example 2.

Instance of Q5: *Why is what the planner propose to sensing* Path1 *first more efficient/cheap than sensing* Path2*?*. The explanation to this question is associated with the total navigation time for Husky1 in each case: if Husky1 takes Path-1, time is 13 mins and for Path-2 the time is 44 mins. This is an example of how the branches can provide foils, which might form the basis for contrastive explanations. The answer to this question can consider the plan metric, which guides the plan solution optimisation.

**Explaining Replanning Queries.** Questions of this nature are associated with planning execution. Visualisation tools such as (Magnaguagno et al. 2017; Cashmore et al. 2019) can be helpful to explain the system behaviour in multiple replanning situations. (Fox, Long, and Magazzeni 2017) describes a way to explain replanning by applying filtering over the set of preconditions required for an action. If all preconditions are achieved, replanning is not needed. This analysis is relevant to our work. However, we are also interested in explaining the best time to replan if the plan presents conditional branches. An instance of *Q6* associated with Example 2 is described here.

Instance of Q6: *Why does the planner need to replan if the outcome of all* sense-path *actions is false?*. The answer to this question is attached to the knowledge acquired by implementing the sensing action. In this example (see Figure 9), if Path-1 and Path-2 are not free the plan cannot be completed. Therefore, the answer to the query should state that none of the possible outputs of the sensing action became available at the planning time. The replanning can be caused due to the noise introduced by the sensory system during plan execution which might prevent the acquisition of the current path's state. The reasoning around these questions should consider the knowledge updates required after executing a $\delta$. The answer to this question links to the runtime plan execution as the state of the paths is known during the plan implementation.

## 7   Conclusions

We have introduced Explainable Planning for Temporally-Contingent Problems (XAIP-TCPs), as a contribution to the Explainable AI (XAI) challenge. The approach was evaluated on a new set of domains, motivated by real-world problems. We define a set of interesting questions from the temporal and contingent planning point of view that covers (i) temporal reasoning, such as timed initial literals and deadlines; (ii) resources, using numerical fluents; (iii) and contingent branches, offering more powerful modelling of mission scenarios. We provide an analysis of the main elements required to deliver effective explanations. We obtained initial results that can lead to additional alternatives of reasoning around plan outputs. The work provides the opportunity to interact with multiple planning choices at the planning and execution time considering the contingency component of our problem solutions. Future work will explore ways to define a good metric for explanation that considers the main characteristics of these problems.

## Acknowledgments

This work was funded and supported by the ORCA Hub (`orcahub.org`), under EPSRC grant EP/R026173/1.

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
