# OpenReview forum: "Explaining Temporal Plans with Incomplete Knowledge and Sensing Information"
_icaps-conference.org/ICAPS/2021/Workshop/XAIP — XAIP 2021_

### Official Review · AnonReviewer2 · 2021-07-04
**Well-written paper. Can extend complexity analysis.**

**Rating:** 8
**Confidence:** 4

**Review:**

## Summary
This paper analyses the explainability of temporal and contingent planning problems for settings with noisy sensing and incomplete knowledge. The work defines what a temporally-contingent planning problem is, and what its solution can look like. It then defines the notion of explainable planning for such problems and discusses possible questions and answers for these settings.

## Feedback

The paper is well-written, easy to read,  and relevant to the XAIP community. The family of XAIP-TC Problems extends the current notions of explainable planning to domains with numeric, temporal, and contingent features. The authors comprehensively discuss the type of questions and answers that might be needed to facilitate the explainable planning for such domains. In my opinion, the discussion of complexity in section 5 can be extended as the current discussion seems informal. More importantly, what will be the complexity of reasoning generated by the AI planner? Also, planning with $K+$ propositions seems computationally hard. The work can include a formal discussion of this.

Minor edits:
1. At many places I think \citet{} would be a better choice to use instead of \cite{}. E.g., section 1, para3, line1; section 1, para4, line1; etc.
2. Def. 1, last sentence. It is not clear at this point what is p in TIL. It becomes clear later, but this def. is incomplete without it.
3. Def. 1 does not talk about $\mathcal{P}_c$ or $\mathcal{P}_t$ as mentioned in the paragraph just before Def. 1.
4. Use \emph{eff} in math mode instead of $eff$. E.g., Def. 2.
5. “AI solver” is used on page 3 (Domain 2’s description) directly, with no reference to it before this point.
6. What is $\delta_d$ in Def. 4? Seems to be a typo.
7. The paper consistently used the phrase “TCP problem”. I think it should either be “TC problem” or just “TCP”.
8. Text in Fig. 1 is difficult to read.
9. Page 4, left side, last para: “The second example is …. is a fixed set”, and “ The second question’s … in each branch”. These sentences seem to be incomplete/incorrect.

---

> ### Author Response · Authors · 2021-07-29
> **Answer to reviewer's comments**
>
> The authors would like to thank the reviewer for the interest in our work, the time invested in our paper, and the helpful comments that will greatly help improve the last version of our manuscript.
>
> The TCP is a special case of Planning in a Partially Observable environment with  Sensing actions  (PPOS)  problems where the source of uncertainty comes from the sensing actions [1]. Our work considers PPOS problems with durative actions. Our problems are simple PPOS problems [2], which can be mapped into Fully Observable NonDeterministic (FOND) planning problems [1]. For a simple PPOS problem, (i) the non-unary clauses in the initial state $I$ are invariant and (ii) no hidden literal emerges in the effects of a nondeterministic action. Another interesting feature of this type of problem is that uncertainty decreases monotonically, i.e., the unknown properties cannot become unknown again after becoming known. The TCP model encloses the uncertainty associated with the nondeterministic action outcomes. The implementation of \textit{belief tracking}---computation of the successor belief state b'---is required to solve the Ptc. The TCP model answers two questions that compile away the uncertainty while searching for a plan.
>
> We recognise numerous questions arise around the model and plan definition. However, this analysis is out of the scope of this paper, which attempts to focus on the explainability side of the TCP solutions.  For more details about the AI planner and the complexity of the system please check our upcoming paper [3].
>
> Regarding minor comments, we fixed all of them in the actual version of the paper.
>
>
> [1] Muise,  C.;  Belle,  V.;  and  McIlraith,  S.  2014.Comput-ing contingent plans via fully observable non-deterministic planning. In AAAI, volume 28.
>
> [2] Bonet, B.; and Geffner, H. 2011.  Planning under partial observability by classical replanning: Theory and experiments.InIJCAI.
>
> [3]Carreno, Y.; Petillot, Y.; and Petrick, R. P. 2021.   Compiling Contingent Planning into Temporal Planning for RobustAUV Deployments. InICAPS Workshop on PlanRob.

---

### Official Review · AnonReviewer1 · 2021-07-06
**Review for "Explaining Temporal Plans with Incomplete Knowledge and Sensing Information"**

**Rating:** 6
**Confidence:** 3

**Review:**

### Summary
This paper provided the definition of the temporally-contingent planning (TCP) problem together with some concrete planning domains modeled in terms of this framework. On top of that, the authors pointed out several directions toward how to provide explanations about some important questions that a user may raise in the context of TCP.
###
I found this paper is easy to read and follow in general. However, there are some concepts that are not clearly specified, which may result in some reader being confused. Apart from this, the remaining part is sound. Thus, from my perspective, I think this paper can be accepted.

### Detailed Comments
(1) In the definition of the temporally-contingent planning problem, the authors said that the problem $P_{tc}$ is defined as $P_{t} \cup P_{c}$, but the formal definitions of $P_{t}$ and $P_{c}$ are not given.
(2) In Definition 2, the set of physical actions consists of instantaneous actions and duration actions which both are the tuple $(a_{pre}, a_{eff}, a_{dur})$. Based upon this definition, it seems that there is no difference between an instantaneous action and a duration action. Moreover, the authors said that $a_{dur}$ is a set of duration constraints, but they did not clarify what are duration constraints. I guess $a_{dur}$ is a set of variables which must hold when the action is executing?

Since those definitions play an important role in this paper, I think it would be better if the authors can clarify these concepts.

---

> ### Author Response · Authors · 2021-07-29
> **Answer for the reviewer**
>
> The authors would like to thank the reviewer for the interest in our work, the time invested in our paper, and the helpful comments that will greatly help improve the last version of our manuscript.
>
> The reviewer is right about the definition of the temporally-contingent planning problem Ptc. The current version of this paper (available here) fixed these issues. The TCP is certainly a combination of the temporal and contingent planning problems; however, considering this paper focuses on the explainability aspects of these problems and their solutions, we attempted to reduce the model definition section.
>
> Regarding Definition2, the tuples defining the instantaneous and physical actions are certainly different (please see the updated version of the paper). The reviewer is right; the instantaneous actions do not have the duration element in the tuple, just the preconditions and effects. Finally, we refer to the "duration constraints" to specify the action duration (reviewer can find similar reasoning in the definition for durative actions in PDDL2.1).

---

### Meta-Review · Area_Chairs · 2021-07-07

**Recommendation:** Accept
**Confidence:** 5

**Metareview:**

We thank the authors for their contribution. It’ll be a great addition to the workshop program.

Please refer to the feedback provided by the reviewers when creating your camera-ready version. Particularly, to improve the clarity of the paper, consider formally defining important terms/notions used throughout the paper, such as the problem $P_{tc}$, which is the main consideration in this paper. It will also be interesting to consider expanding the discussion in Section 5, as suggested by reviewer 2, to include some analysis on the practical aspect of this work (i.e., what is the overall complexity involved in finding solutions for some of the queries considered in this work).

We are looking forward to an interesting and fruitful discussion at the workshop.

---

### Decision · Program_Chairs · 2021-07-08

Accept